# Urinary Biomarkers and Point-of-Care Urinalysis Devices for Early Diagnosis and Management of Disease: A Review

**DOI:** 10.3390/biomedicines11041051

**Published:** 2023-03-29

**Authors:** Beatriz Sequeira-Antunes, Hugo Alexandre Ferreira

**Affiliations:** 1Institute of Biophysics and Biomedical Engineering, Faculty of Sciences, University of Lisbon, Campo Grande, 1749-016 Lisboa, Portugal; 2Exotictarget, 4900-378 Viana do Castelo, Portugal

**Keywords:** urine metabolites, biomarkers, biosensing, microfluidics, urinalysis, continuous monitoring, noninvasive health monitoring

## Abstract

Biosensing and microfluidics technologies are transforming diagnostic medicine by accurately detecting biomolecules in biological samples. Urine is a promising biological fluid for diagnostics due to its noninvasive collection and wide range of diagnostic biomarkers. Point-of-care urinalysis, which integrates biosensing and microfluidics, has the potential to bring affordable and rapid diagnostics into the home to continuing monitoring, but challenges still remain. As such, this review aims to provide an overview of biomarkers that are or could be used to diagnose and monitor diseases, including cancer, cardiovascular diseases, kidney diseases, and neurodegenerative disorders, such as Alzheimer’s disease. Additionally, the different materials and techniques for the fabrication of microfluidic structures along with the biosensing technologies often used to detect and quantify biological molecules and organisms are reviewed. Ultimately, this review discusses the current state of point-of-care urinalysis devices and highlights the potential of these technologies to improve patient outcomes. Traditional point-of-care urinalysis devices require the manual collection of urine, which may be unpleasant, cumbersome, or prone to errors. To overcome this issue, the toilet itself can be used as an alternative specimen collection and urinalysis device. This review then presents several smart toilet systems and incorporated sanitary devices for this purpose.

## 1. Introduction

Health monitoring helps in identifying signs associated with diseases, making it an essential element for the diagnosis, treatment, and rehabilitation of severe medical conditions [1]. Such monitoring on a continuous basis can help provide for the early detection of diseases, which in turn can lead to improved health outcomes for patients by enabling the early start of treatment, as well as by reducing both suffering and medical costs [2].

There is currently a revolution taking place in the health sector, where traditional medicine is being replaced with a proactive and person-centred medicine, so-called precision medicine [3]. This change is possible due to technological advances, witnessed over the last few years, as they allow the continuous recording of large volumes of clinical data, thus improving the health of an individual [4].

One of the goals in clinical practice is to gather diagnostic data with minimal discomfort and invasiveness for patients. Therefore, tests that assess biomarkers found in biofluids such as blood or urine are highly sought after [5].

The main objective of biomarker discovery is identifying protein markers that can improve early diagnosis. The majority of the studies used blood serum and/or plasma to identify possible biomarkers, as blood is generally regarded as an ideal fluid for the evaluation of systemic diseases [6]. However, due to the invasive collection process, blood is not suitable for routine monitoring [3].

Another biofluid traditionally used is urine. Urine serves as a more promising sample for ubiquitous health monitoring [3]. Unlike blood, urine collection is totally free of contact with the body, is painless, noninvasive, and does not cause any physical discomfort [7]. Another advantage of using urine is that there is an abundance of urine-based biomarkers for specific conditions, particularly cancer and kidney-related issues [4].

In general, urine is an accessible sample type that can be collected non-invasively and processed with relative ease. However, the use of urine as a source of biomarkers creates challenges due to its complex composition and inter-individual variability [8]. In addition, some biomarkers in urine may be present at low concentrations, requiring additional processing steps to increase their detectability. Another issue with urine as a sample source is the risk of contamination during collection or handling [9], which can affect the accuracy of biomarker detection. To address these challenges, ultra-sensitive biosensing and microfluidic technologies offer promising solutions. These technologies can minimize sample handling and processing steps, reducing the risk of contamination and the loss of biomarkers, while improving the accuracy and precision of detection [10,11]. As a result, these technologies have the potential to overcome the limitations associated with urine as a source of biomarkers and provide valuable insights for biomedical research and clinical practice.

In the last decade, there has been a significant increase in the adoption of mobile and wearable devices, along with the introduction of point-of-care diagnostic devices using microfluidics technology that offer the possibility of continuous monitoring. Microfluidics involves the science and technology of the manipulation of biofluids at the micro- to nano-liter level, which can potentially indicate a more efficient approach, with the use of minute volume samples, the reduced consumption of reagents, and a lower environmental footprint at disposal, whilst being affordable [4,12].

In this review paper, the potential of combining urine biomarkers, biosensing technologies, microfluidics, and continuous monitoring devices will be explored for the early detection and personalised management of chronic diseases. This paper also aims to highlight the latest developments in the field and to discuss the related opportunities and challenges.

## 2. Urine Metabolites and Their Role as Biomarkers

According to the World Health Organization (WHO), biomarkers are defined as “any substance, structure, or process that can be measured in the body or its products and influence or predict the incidence of outcome or disease” [13]. One of the many advantages of using biomarkers is that, unlike disease symptoms, which are subjective, biomarkers provide an objective and measurable way to characterise the disease [14]. They can often be measured by analysing blood or urine samples, helping clinicians avoid complex invasive procedures [15].

Urine is a rich source of cellular metabolites and an important, easily accessible biological fluid, and one of the most useful biofluids for routine testing [16]. Metabolomics is a novel field of science that seeks to quantitatively describe the fluctuations of numerous metabolites within organisms. It offers significant benefits in identifying disease biomarkers because certain metabolites can vary among individuals, indicating their unique metabolic traits and the underlying manifestations of their disease [17,18,19]. However, the normal reference values of various urine metabolites have not been established yet, and further clinical validation is necessary.

Metabolomic studies typically begin with sampling, followed by sample analysis. There are several techniques to do this analysis, although the most common used is nuclear magnetic resonance spectroscopy (NMR) [20,21]. This technique is non-destructive in nature, quantitative, and has a safe metabolite identification that provides detailed information on the molecular structure. Other techniques are used, such as gas chromatography mass spectrometry (GC-MS), liquid chromatography mass spectrometry (LC-MS), and the enzyme-linked immunosorbent assay (ELISA) [20].

Currently, approximately 4500 metabolites have been documented in urine, showing connections to approximately 600 human conditions including, for example, obesity, cancer, inflammation, and neurological diseases [7].

### 2.1. Arterial Hypertension

Hypertension, which is characterized by elevated blood pressure in systemic circulation arteries, is among the most widespread of chronic diseases [19,22]. According to statistics, the global prevalence of hypertension was 26.4% by 2000, and it is anticipated to climb to 29.2% by 2025 [19]. This disease can lead to other several diseases, such as stroke, heart disease, and kidney failure [22]. In this way, it can be said that hypertension is a key modifiable risk factor for cardiovascular morbidity and mortality [23] for which it is important to find urinary biomarkers that would allow for an early diagnosis.

Previous metabolomic studies on hypertension have been mainly based on blood and urine samples. The study carried out by Loo et al. [24] had as the identification of a panel of urinary metabolites whose changes are related to risk factors of cardiovascular disease (CVD) as its main objective. The results indicated that there are six urinary metabolites associated with blood pressure (proline-betaine, carnitine, hippurate, 4-cresyl sulphate, phenylacetylglutamine, and *N*-methyl-2-pyridone-5-carboxamide).

In recent years, trimethylamine-*N*-oxide (TMAO) has also emerged as a potential biomarker for the development of CVD [25,26,27]. This metabolite is a small, organic, gut microbiome-generated compound whose concentration increases after ingesting dietary L-carnitine and phosphatidylcholine rich foods such as red meat, eggs, and fish [27].

### 2.2. Oxidative Stress and Inflammatory Disease

Oxidative stress is caused by a production imbalance between reactive oxygen species (ROS) and antioxidant defences [28,29]. That phenomenon, if uncontrolled, initializes numerous intracellular signaling pathways that trigger apoptosis or excessive cell growth, which can lead to organ dysfunction in the heart, pancreas, kidneys, and lungs. As a consequence, hypertension, diabetes, chronic kidney disease, and pulmonary disorders can develop [30].

Inflammation is a natural defence mechanism of the immune system that can be triggered by a variety of factors [31]. Oxidative stress has the ability to activate several transcription factors that cause specific genes involved in inflammatory pathways to be expressed differentially [30]. Therefore, there is evidence that oxidative stress and inflammation are coincident phenomena that exert an influence on each other [32].

If the level of ROS exceeds the antioxidant capacity of a cell, cellular biomolecules such as DNA, lipids, and proteins are oxidized. This leads to the creation of compounds that can serve as biomarkers of oxidative stress. The most commonly used urinary oxidative stress biomarkers are 8-hydroxy-2′-deoxyguanosine (8-OHdG) [30,33,34], phenylacetylglutamine, adenine, glycine [35], lactate [21], 8-isoprostane [33,36], malondialdehyde (MDA), F2-isoprostanes, and dityrosine (diY) [34].

### 2.3. Chronic Kidney Disease

Chronic kidney disease (CKD) is a condition that occurs when the kidney gradually loses its function or when a glomerular filtration rate (GFR) is less than 60 mL/min per 1.73 m^2^ for 3 or more months, regardless of the cause [37]. The prevalence of CKD is increasing worldwide, and the mortality rate continues to be unacceptably high [38]. This disease is a complex one because can it affect multiple organ systems and often coexists with numerous associated conditions, such as CVD, diabetes mellitus, and chronic inflammation [39]. GFR estimation and albuminuria are commonly utilized to diagnose and predict the prognosis of CKD in clinical practice. GFR estimation correlates with the severity of kidney malfunction, whereas albuminuria indicates the existence of kidney damage [38].

Detecting CKD early is a crucial and unsatisfied medical requirement, not only for forecasting and impeding CKD progression, but also for enhancing patient survival and decreasing associated morbidities. That can be accomplished through the identification of suitable biomarkers [38]. Some new possible biomarkers of the diagnosis of CKD and its prediction of outcomes have been identified, such as creatinine [39,40].

### 2.4. Urinary Tract Infection

Urinary tract infections (UTI) affect 150 million people each year worldwide, with an annual incidence of 12.6% in women and 3% in men [41]. It appears due to microbial pathogens invading the urinary tract, which can lead to several clinical manifestations [42].

Currently, urine culture is the standard method in the diagnostics of UTI [43]. However, this method is slow, and therefore the diagnosis happens with a considerable delay, which is not desirable [42]. Also, urine test strips can be used for rapid UTI screening. These strips evaluate several biomarkers, although only nitrite and leukocyte esterase show an independent relation with UTI. Nonetheless, there are reports of considerable variations in the test performances, which limit their further use, and as such new biomarkers that allow for a quick diagnosis are warranted [43]. Several promising urine biomarkers of UTI such as agmatine and N6-methyladenine have already been identified [44].

### 2.5. Alzheimer’s Disease

Alzheimer’s disease (AD) is a progressive neurodegenerative disease and the leading cause of dementia, the incidence of which is rapidly increasing [45]. AD is characterised by the formation of extracellular amyloid plaques which result from the accumulation of amyloid β proteins, and intracellular neurofibrillary tangles due to the aggregation of tau proteins, leading to neuronal signalling disruption and cell death [46,47]. Thus, the development of the disease involves numerous factors such as amyloid β build-up, oxidative stress, tau phosphorylation, lipid imbalance, mitochondrial dysfunction, and inflammation [48].

AD is an ailment that frequently impacts the elderly population, and its symptoms are discernible only in the advanced stages, rendering early diagnosis a challenging task [49]. Currently, diagnosis is based on subjective neuropsychological tests and by late-stage biomarkers in cerebrospinal fluid (CSF), which require a lengthy and often a painful procedure [47]. Other often used biomarkers include imaging biomarkers such as the uptake of the ^11^C-Pittsburgh compound B (^11^C-Pib), as assessed using positron emission tomography (PET), which binds to amyloid plaques, and the atrophy of the hippocampus and mesial temporal structures, as assessed by magnetic resonance imaging (MRI) [50,51]. Nonetheless, these imaging modalities and examination procedures can also be lengthy and very expensive, whilst also using ionising radiation (PET) and high magnetic fields (MRI), which present risks to the patient [52].

The signs of AD pathology may also be found in the urine, so it is important to know which biomarkers can be used. Due to the oscillation of urinary flow, normalisation of the biomarker concentration is recommended in most cases. This is easily achieved through the relationship between the concentration of the biomarker and the concentration of creatinine [45]. Examples of this kind of biomarker are amino acid-conjugated acrolein (AC-Acro/Cre) and 3-hydroxypropyl mercapturic acid (3-HPMA/Cre) [53,54]. Other urinary biomarkers can be associated with AD, such as 8-OHdG [45,49], and knowing the relationship between AD and oxidative stress, 8-isoprostane and glycine can be included as AD biomarkers [45].

### 2.6. Oncologic Diseases

Cancer is one of the major causes of mortality worldwide, and according to the America Cancer Society, in 2022 there will be an estimated 1.9 million new cancer cases diagnosed and 609,360 cancer deaths in the United States [55]. Its incidence is predicted to increase significantly, with a forecast of 22 million new cancer cases and 13 million cancer-related deaths occurring annually by 2030 [56]. The best chance of reducing these numbers is through early detection. To this end, the use of biomarkers can be useful. However, in recent years the identification of novel biomarkers in biological fluids has increased significantly, although further validation is needed.

#### 2.6.1. Lung Cancer

Lung cancer (LC) has a high mortality rate globally, and in most cases, diagnosis is often made at a late stage when the process of metastization has already begun [57]. To avoid such a scenario, the use of biomarkers may be useful. Some examples of recurrent biomarkers proposed in scientific papers investigating lung cancer were tyrosine and tryptophan [58], hippurate [59,60,61], *N*-aceglutamide, β-hydroxyisovaleric acid, α-hydroxyisobutyric acid, and creatinine [59], valine, proline betaine, taurine [61], and phenylalanine [58,61].

#### 2.6.2. Breast Cancer

Breast cancer (BC) is the second most common cancer overall and the most frequent type of cancer in women worldwide [62]. The early diagnosis of breast cancer greatly increases the chance of cure and survival from the disease. Mammography is the gold standard for BC screening; however, it has limited sensitivity, involves exposure to ionising radiation, and it has not been shown to significantly contribute to decrease mortality [60,63]. In this way, noninvasive tests for BC with high sensitivity are needed. The use of urinary metabolomics for breast cancer detection at an early stage has increased over the last few years [60]. The major contributing metabolites identified were 8-OHdG, 1-methyladenoside, 1-methylguanosine [64,65,66,67], creatinine, succinate, valine, and isoleucine [60,68]. Nam et al. also identified homovanillate, 4-hydroxyphenylacetate, 5-hydroxyindoleacetate, and urea in urine as biomarkers of BC [69].

#### 2.6.3. Bladder Cancer

Bladder cancer is the seventh most common cancer, with an average of 356,000 new cases diagnosed worldwide every year [56,70]. It is the second most prevalent malignancy in middle age and in elderly men after prostate cancer [71]. The current standard procedure for bladder cancer detection and monitoring tumour progression and recurrence involves urine cytology, cystoscopy, and biopsy; however, these techniques have a number of limitations, as low sensitivity, in addition to the fact that is expensive, invasive, and painful [56]. Thus, new diagnostic approaches that improve the diagnostic accuracy, reduce pain levels and that are noninvasive are needed.

Screening bladder cancer patients through urine metabolomics biomarker technology is a promising strategy to improve detection and diagnosis [72]. Some of the commonly used biomarkers for bladder cancer include hippurate [70], lactate [73], succinate [73,74], phenylalanine, tyrosine, tryptophan, leucine, uric acid [71], carnitine [72,74], and taurine [70,73].

#### 2.6.4. Prostate Cancer

Prostate cancer is among the most prevalent types of cancer in men across Europe, and its occurrence has surged dramatically over the last twenty years [75]. At present, the detection of prostate cancer is still an imprecise practice. The screening process involves measuring the level of prostate-specific antigen (PSA) in the blood and performing a digital rectal examination [76]. PSA is the only biomarker commonly utilized in the diagnosis of prostate cancer patients; however, its sensitivity and specificity are inadequate, resulting in the occurrence of false-negative and false-positive test outcomes [77]. To reduce incorrect results and increase the accuracy of diagnosis, it is necessary to search for additional prostate cancer biomarkers.

Sreekumar et al. [78] reported that increasing levels of proline, kynurenine, uracil, and glycerol-3-phosphate were significantly correlated with disease progression. The authors also found that sarcosine concentrations increased in patients with prostate cancer [78]. Some studies confirmed the potential of sarcosine as a noninvasive screening tool for prostate cancer [75,77,79,80,81,82,83]. Other identified contributing metabolites identified were leucine [81,83], creatinine [77,79,83], tyrosine, tryptophan, taurine [83,84], and alanine [76,81].

#### 2.6.5. Gastric Cancer

Gastric cancer was estimated to be responsible for over one million new cases in 2018 worldwide and for more than 700,000 deaths [56]. Currently, the standard diagnostic method for gastric cancer is gastroduodenal endoscopy; however, this technique has several drawbacks, such as invasiveness and high-cost [85]. Currently, no diagnostic method is perfect for detecting gastric cancer at an early stage since, in most cases, it is asymptomatic [20]. In recent years, several urinary biomarkers have been identified as new tools for the early screening of gastric cancer. Among the 25 metabolites investigated by Chan et al. [86], only 2-hydroxyisobutyrate, 3-indoxylsufate and alanine provided useful information for gastric cancer diagnosis. Another study conducted by Dong et al. in 2009 [87] concluded that the level of urinary prostaglandin E2 metabolite (PGE-M) was higher in gastric cancer patients than in a control group. Furthermore, arginine, leucine, isoleucine, valine, citric acid, succinate, histidine, methionine, serine, aspartate, taurine, tyrosine, lactate, and phenylalanine [85,88] have been proposed as biomarkers for gastric cancer.

#### 2.6.6. Kidney Cancer

Kidney or renal cancer is among the top ten most prevalent forms of cancer, and it is more frequent in men than in women [56]. According to the American Cancer Society’s projection for 2022, roughly 79,000 new cases of kidney cancer and approximately 13,920 fatalities due to this type of cancer were anticipated [89]. Currently, most kidney cancers are detected before symptoms appear, namely when performing routine examinations, as well as when investigating symptoms such as back or abdominal pain using imaging [90]. However, it remains true that alternative diagnosis methods, particularly those that use urine markers, could be useful in detecting kidney cancer even earlier. Therefore, the identification of a screening biomarker has the potential for substantial health benefits. Promising biomarkers include acylcarnites such as isobrutyrycarnite, suberoylcarnite, and acetylcarnite [91].

## 3. Urine Proteins Biomarkers

Apart from metabolites, the presence or absence of proteins in urine can provide valuable information about various medical conditions, including kidney injury and certain types of cancers. Table 1 provides a summary of the proteins used as biomarkers for various conditions, along with their normal level values in urine.

One of the most common urine proteins used as biomarker is urinary albumin, which is produced in the liver and helps to maintain the balance of fluids in the body. Overall, urinary albumin can be a useful biomarker for a range of diseases and conditions that affect the cardiovascular system, such as hypertension [23,92], the liver [93], and the kidneys, such as chronic kidney disease [38].

Although urinary albumin is useful for assessing kidney function, it is important to look for other indicators of kidney injury other than GFR. Current established filtration markers for the prediction of outcomes in patients with CKD include creatinine and cystatin C [40,94]. Other biomarkers potentially useful to indicate kidney damage are β2-Microglobulin (B2M) and Beta Trace Protein (BTP). These proteins are filtered out of the blood by the kidneys and excreted in the urine, normally in small amounts of less than 20 and 300 micrograms per liter (µg/L), respectively. Thus, elevated levels of BTP and B2M in urine can be an indication of kidney damage or dysfunction, and can be used as biomarkers to diagnose and monitor various kidney-related diseases and conditions [38,95].

Uromodulin is a protein that is produced by the kidney and is the most abundant protein found in urine [96]. This protein can be used as a biomarker, since its concentration gradually decreases with worsening kidney function, so in patients with CKD, its concentration will be lower [38]. Moreover, the Neutrophil Gelatinase-Associated Lipocalin (NGAL) [94,96] and Kidney Injury Disease-1 (KIM-1) [96,97] have also been suggested to be potential biomarkers of CKD.

Inflammation and CKD are closely related, since inflammation can lead to damage of the kidney tissue or exacerbate existing kidney damage and hasten the progression of the disease. Therefore, the study and identification of related biomarkers is essential in the prevention and management of kidney disease. Inflammatory biomarkers such as C-Reactive Protein (CRP) [98], Interleukin-6 (IL-6) [98,99], Tumor Necrosis Factor-alpha (TNF-α)[98,99,100], and Growth Differentiation Factor-15 (GDF-15) [98,101] have been linked to renal function decline. Regarding urinary tract infections, some proteins have been identified as potential biomarkers, such as lactoferrin (LF) [102], xanthine oxidase (XO), and myeloperoxidase (MPO) [43].

Urine protein biomarkers have also shown promise in the detection and monitoring of certain types of cancer. These biomarkers are often produced by cancer cells or by the body in response to cancer. An example is Prostate-Specific Antigen (PSA), which is a protein produced by the prostate gland, whose elevated urinary levels may indicate prostate cancer [79]. Other biomarkers that are being studied for their potential use in the detection and monitoring of cancer include Human Epididymis Protein 4 (HE4) for ovarian cancer [103], Bladder Tumor Antigen (BTA) for bladder cancer [104], and Matrix Metalloproteinases (MMPs) for various types of cancer, such as MMP-9 for breast cancer [105].

The diagnosis and monitoring of neurodegenerative diseases, such as Alzheimer’s disease, using urine biomarkers have been studied over time. One such biomarker is the Beta-Amyloid (βA) protein, which is detected in the urine. Beta-amyloid is a protein that is present in the brains of individuals with Alzheimer’s disease, and may also be present in the urine of these individuals [106].

Other urinary biomarkers can be associated with AD, such as AD-associated Neuronal Thread Protein (AD7c-NTP) [107], osteopontin, gelsolin, and Insulin-like Growth Factor-Binding Protein 7 (IGF BP7) [46].

**Table 1 biomedicines-11-01051-t001:** Summary of the protein biomarkers identified and their normal urinary values.

Disease and Condition		Protein Biomarker	Normal Urinary Levels	Reference
Chronic Kidney Disease		Albumin	<30 mg/g of creatinine	[108]
		Creatinine	0.56–2.26 g/L (Men) 0.40–1.74 mg/dL (Woman)	[109]
		Cystatin C	<100 µg/L	[110]
		B2M	<160 µg/L	[108]
		BTM	600–1200 µg/L	[108]
		Uromodulin	100 mg/day	[111]
		NGAL	<50 µg/L	[108]
		KIM-1	<1 ng/mL	[112]
Inflammatory Disease		CRP	<6 mg/L	[113]
		IL-6	0.7–4.1 ng/L	[114]
		TNF-α	<1.3 pg/mL	[115]
		GDF-15	1.2–4.6 µg/g of creatinine	[116]
Urinary Tract Infection		LF	30.4 ± 2.7 ng/mL	[102]
		XO	104.57 ± 49.28 U/L	[117]
		MPO	414.09 ± 93.31 U/L	[117]
Cancer	Prostate	PSA	<4 ng/mL	[118]
	Ovarian	HE4	<78.6 pmol/L (premenopausal) <122.5 pmol/L (posmenopausal)	[119]
	Bladder	BTA	<14 U/mL	[120]
	Breast	MMP-9	Not detectable	[121]
Alzheimer’s Disease		βA	0.003–1.11 ng/mL	[106]
		AD7C-NPT	0.04–2.07 ng/mL	[122]
		Osteopontin	4 mg/day	[111]
		Gelsolin	1000–1200 pg/mg total protein	[123]
		SPP1	12–18 ng/mg total protein	[123]
		IGF BP7	4.8–5.2 pg/mg total protein	[123]

## 4. Urine Nucleic Acids as Biomarkers

In addition to hundreds of proteins, urine also contains exfoliated tumor cells and tumor cell-free amino acids, in addition to tumor-derived DNA, mRNA, and microRNA (miRNA) [124,125].

In recent years, there has been growing interest in the use of urinary methylation-based biomarkers for the diagnosis and monitoring of some types of urogenital cancers, particularly in their early stages, as alterations in DNA methylation are thought to be among the earliest events in the development of tumors. According to Bryzgunova et al., methylation of Glutathione S-Transferase P1 (GSTP1) shows potential as a promising biomarker for prostate cancer, which can be detected in urine samples from affected patients [126].

Apart from alterations in DNA methylation, messenger RNA (mRNA) molecules in urine can be used as biomarkers. An example of that is the two available tests, SelectMDx and ExoDx, for the detection of prostate cancer. The former detects Distal-Less Homeobox 1 (DLX1) and Homeobox C6 (HOXC6) mRNA in urine after prostate massage, while the latter detects Prostate Cancer Antigen 3 (PCA3), Erythroblast Transformation-Specific (ETS)-related gene (ERG) and Sterile alpha Motifpointed Domain-Containing ETS transcription Factor (SPDEF) in urinary exosomes and does not require a digital rectal exam [127,128].

Feng et al. investigated the potential use of C-C Motif Chemokine Ligand 5 (CCL5) and C-X-C Motif Chemokine Ligand 1 (CXCL1) mRNA levels in urinary sediment as prognostic biomarkers for diabetic nephropathy. The results showed that both CCL5 and CXCL1 were upregulated in diabetic nephropathy patients and were associated with a decline in renal function [129]. In addition, low levels of CD2-associated protein (CD2AP) mRNA in urinary exosomes were associated with an increased risk of kidney disease [130].

The concept of liquid biopsy, which involves the detection, analysis, and monitoring of cancer through various bodily fluids such as urine [131], was initially introduced for circulating tumor cells but has since been extended to include circulating tumor DNA [132]. Circulating tumor DNA has been studied as a potential biomarker for various types of cancer, with a particular focus on genitourinary tract cancers, such as bladder cancer. Regarding bladder cancer, Christensen et al. were able to detect urinary cell-free DNA, specifically targeting three hotspot mutations in Phosphatidylinositol-4,5-bisphosphate 3-Kinase Catalytic subunit Alpha, PIK3CA (E545K) and Fibroblast Growth Factor Receptor 3, and FGFR3 (S249C, Y373C) [133]. Other possible biomarkers were identified, such as Long Interspersed Nuclear Element-1 (LINE1) [134] for lung cancer, p53 mutation (codon 249) for hepatic cancer [135], and vimentin hypermethylation for colorectal cancer (Song2012 [136]).

MicroRNA (miRNAs) are small, typically 20–25 nucleotides in length, non-coding RNA molecules that play a crucial role in the regulation of gene expression [125]. miRNAs have emerged as promising biomarkers for the diagnosis and monitoring of various types of cancer. In prostate cancer, several miRNA were identified, such as miR-107, miR-574-3p [137], miR-205, miR-214 [138], and miR-888 [139]. Other miRNA biomarkers that are being studied for their potential use in the detection and monitoring of cancer include miR-144-5p [140], miR-23b/27b [141], and miR-145 [140] for bladder cancer, and miR-96 and miR-214 [142] for urothelial cancer.

Previous studies showed that urinary levels of microRNA correlated with kidney diseases. Lv et al. [143] concluded that miR-29c from urinary exosomes was significantly downregulated in CKD patients. In a study conducted by Szeto et al. [144], miRNA levels were measured in urinary sediment. The researchers observed a correlation between the expression of urinary miR-21 and miR-216a and the rate of decline in renal function, as well as the risk of developing renal failure requiring dialysis [144]. In most cases, kidney diseases are linked with cardiovascular diseases, such as renovascular hypertension, which is high blood pressure caused by renal artery disease. Through their studies, Yang et al. and Know et al. have identified miR-26A [145] and miR-21, miR-93, and miR-200b [146], respectively, as potential markers for diagnosing renovascular hypertension.

Various urine nucleic acid biomarkers described previously are currently under study. Consequently, there are no normal reference values yet and further clinical validation is warranted.

## 5. Biosensing Technologies and Approaches

In the last years, the use of biosensors for medical applications has increased, since these devices can detect specific biological analytes and monitor their functions within a biological environment [147]. These devices use biological components, such as enzymes, antibodies, nucleic acids, or cells, in combination with transducers to detect and quantify the presence of a specific analyte, as illustrated in Figure 1. The biological component of the biosensor recognises and interacts with the target analyte, resulting in a measurable signal that is transduced into an electrical, optical, or another type of signal by the transducer [10].

Biosensing offers several advantages over traditional analytical methods, including rapid and real-time analysis, minimal sample preparation, and the ability to perform on-site or in-field measurements. However, there are some limitations and challenges associated with biosensors, including non-specific binding, interference from the surrounding environment, and fragility [10,147,148]. Table 2 lists some transducer mechanisms used in biosensing and their advantages and disadvantages.

Recent advances in biosensor design and fabrication have led to the development of more sensitive, selective, and rapid biosensors with improved detection limits, response times, and specificity [148]. These advances have been made possible by new materials and fabrication techniques, as well as by the integration of nanotechnology and microfluidics. Overall, the field of biosensors is expected to play an increasingly important role in a wide range of applications in the future.

## 6. Microfluidics Technologies and Approaches

Microfluidics is the science and technology of systems that manipulate fluids at the submillimeter scale within microchannels or other microstructures [149]. It is considered a multidisciplinary technology that links several different sciences including chemistry, biochemistry, engineering, physics, and micro- and nano-technology [10].

Certain properties of microfluidics technologies, such as the ability to use very small quantities of samples and reagents, reduced cost and waste generation, high resolution and sensitivity, and faster reaction times and analysis have led to these technologies being extensively studied and developed over the last decades [150,151,152].

To design and fabricate microfluidics platforms, it is necessary to consider several factors, such as the materials and fabrication methods to be used.

The materials most used for microfluidics structures are glass and silicon. These materials were useful to guarantee the stability of the geometry of the microstructures; however, they were expensive and required high costs of fabrication. Furthermore, unlike glass, silicone is opaque, which, makes it not useable with conventional optical methods of detection [151,152]. Various kinds of polymers with different properties, such as polymethylmethacrylate (PMMA) and polydimethylsiloxane (PDMS), have more recently been chosen to fabricate microfluidics. Compared to glass and silicon, these polymers are cheaper. Furthermore, they have a satisfactory optical transparency and mechanical stability, and can be easily replicated and bonded to a diverse range of substrates [152]. These characteristics make PMMA and PDMS the materials of choice for the prototyping of microfluidics devices. Currently, even thermoplastics and paper are accepted as fabrication materials [150]. Table 3 summarises the main advantages and limitations of the different materials used in the fabrication of microfluidic structures and systems.

Regarding the fabrication methods, several different techniques can be considered, such as injection moulding, hot embossing, photolithography, soft lithography, and 3D printing. The first two fabrication methods, injection moulding and hot embossing, use the wide range of available thermoplastics to generate high throughput, cost-efficient, and precise microfluidics [153]. Injection moulding occurs in four main steps. First, the thermoplastic is melted to a liquid state inside of a comprehensible chamber. Next, the two halves of the mould are compressed, creating a mould cavity where, in the next step, the thermoplastic is injected at a specific rate. Finally, the mould is cooled, and the cast pat is removed from the mould [153]. The work process of hot embossing is quite like the one described above; however, instead of injecting the thermoplastic into a cavity, the material is spilled and pressured against the mould to impress the desired features in the softened thermoplastic [154].

Another technique that is used is photolithography. This one consists in drawing, photographing, and reducing the pattern to a negative, called photomask, with the required final size. Then the photomask is projected onto a substrate wafer which has been coated in a photosensitive polymer. This allows the transfer of the generated pattern from the photomask to the substrate wafer. Subsequently, depending on the type of photomask used (inverted or non-inverted) deposition or etching steps follow until the microfluidic structure is complete [153]. Among the different techniques, the soft lithography is one of the most popular methods for fabricating biomedical microfluidics devices [150,154]. This technology, also known as replica moulding, includes some steps. First, the hard master is created, and the liquid polymer is poured inside of it. The mould is then heated through heat-curing, and the polymer is then peeled [154]. Both techniques, photolithography, and soft lithography, can only be carried out in an environment free from airborne particulate or chemical contaminants, meaning these methods must be conducted in a cleanroom, which can be considered a limitation [150].

Currently, researchers are also making use of 3D printers to fabricate microfluidics devices [150]. It is a layer-by-layer manufacturing technology that, even if relatively new, is becoming a successful approach to the fabrication of microfluidic structures and systems [154]. Each of the techniques described above has advantages and disadvantages, as illustrated in Table 4.

Microfluidics devices can be used to provide a rapid and accurate diagnosis since they allow for the analysis of various kinds of samples, such as blood, saliva, or urine. An example of the application of microfluidics devices was demonstrated by Narimani et al. [155], that created a cheap, portable, and efficient method to determine creatinine levels based on synthesised nanoparticles and colorimetric image-processing techniques. Also, Sununta et al. [156] and Fu et al. [157] projected a microfluidic-based device that allowed for the determination of creatinine in urine samples [156] and human serum [157], respectively. These devices were based on the Jaffé reaction between the creatinine and picric acid in alkaline conditions, generating a colorimetric creatinine-alkaline picrate complex. Other substances such as hormones can be analysed with microfluidics devices. A known example is paper-based microfluidics within home pregnancy tests, which detect the urinary concentration of the human chorionic gonadotropin hormone. Choi et al. [158], to monitor hydration and manage health disorders, created a microfluidic platform that captured, stored, and analysed sweat biomarkers and temperature. More complex diseases, such as cancer [159] and genetic-based diseases [160,161], can be monitored using this technology. Nagrath et al. [162] developed the first microfluidic technology to capture circulating tumour cells from whole blood using microposts coated with antibodies. Also, Baldacchini et al. [163] created and used a BioFET immunosensor for the detection of thep53 suppressor in a physiological environment. This microfluidic device includes a MOSFET that is connected to a sensor chip. The baseline work of this device consists of the detection of the variation of gate surface potential, which arises due to the binding of the p53 protein with its antibody present on the surface of the sensor chip [163].

## 7. Point-of-Care Diagnostics: Urinalysis

Urinalysis is performed sporadically every other week, month, three or six months, or every year, depending on the medical condition, such as urinary tract infection, chronic kidney disease, diabetes, or other diseases and health checkups. Typically, a healthcare provider will ask the patient to provide a urine sample, which is then analysed in a laboratory. This collection is performed manually by the patient, which may become a limitation to the procedure [164].

Urinalysis is commonly performed using traditional methods such as urine culture, urine microscopy, and dipstick tests. The dipstick test, in particular, is widely used due to its ease of use, quick response time, and low cost [165]. In this test the dipstick is immersed in the urine sample, which will react with the chemical reagents present in each one of the pads of the plastic strip, causing a colour change that is compared to a colour chart on the dipstick container. The dipstick test has limitations, since interpreting the test results with the naked eye can be subjective, and variations in lighting or individual perception may affect the precision of the colour of the test, leading to inaccuracies. This can be overcome by using biosensing and microfluidics techniques, namely optical biosensors to analyse the dipstick’s colour changes. Regarding manual urine collection, a toilet itself as an alternative specimen collection and urinalysis device can be used, allowing for the continuous assessment of relevant biomarkers, as the urine can be collected and analysed with every use of the toilet.

There are studies that report prototype mountable smart toilets that collect data from urine and stool passage and link these data to individual users. An example of this was the studies conducted by Park et al. [18] and Bae et al. [164], where a retractable dipstick was placed on the toilet to analyse ten biomarkers to perform a urinalysis. Schlebusch et al. [166] presented an “intelligent toilet” that performed an extensive health check. The system measured skin temperature, electrocardiography and bioimpedance, weight and urine turbidity, colour, and glucose content. As with the previous studies, a diagnostic dipstick was immersed into the urine, and the colour change of the dipstick was measured by optical biosensors. Mao et al. [167] created a household prototype that performed a fast urine analysis at home. The device included a flexible biosensor for the detection of glucose, sodium, and potassium ions, and a Bluetooth component that could send the data to the mobile phone, which allowed for the tracking of long-term changes based on daily records.

Instead of dipsticks, incorporated devices can also be used to perform urinalysis. An Israeli medical device startup, Olive Diagnostics [168], has come up with a way to use artificial intelligence (AI) to analyse what is in your toilet bowl. To do that, they create Olive KG, which uses an optical sensor and an AI-powered device that performs a high-quality analysis of important parameters in the patient’s urine upon each urination. The parameters include red blood cells, proteins, pH, nitrite, and features such as volume, pressure, colour, and frequency of urination. This device has received the EU CE marking as medical device and can be mounted on any toilet to generate real-time personal data, which are then sent to the cloud and to the medical assistant.

Another incorporated device is U-Scan from Withings [169]. It is presented as a versatile cartridge-based platform that is able to detect, measure, and analyse multiple biomarkers in urine according to the kind of cartridge, whether it be a U-Scan Cycle Sync or a U-Scan Nutri Balance. The former is a hormonal-based cycle tracking solution and allows for the detection of some biomarkers, such as luteinising hormone, specific gravity, and pH. The latter, the U-Scan Nutri Balance, can seamlessly test several key biomarkers for hydration and nutrition via automatic urine analysis. The microfluidic system is similar to the dipsticks, since the device includes small paper strips whose colour will change according to the reaction performed. These colour changes will then be read by the device’s scanner. According to the manufacturer, this product will be available in Europe at some point in the second trimester of 2023, and later in the USA, as the Food and Drug Administration (FDA) will decide whether to clear the product for consumer use. Figure 2 exemplifies the process of continuous monitoring using a toilet to collect the sample.

## 8. Conclusions and Future Directions

The lack of reliable, easy-to-use diagnostic solutions for point-of-care urinalysis is a significant challenge in healthcare. Traditional diagnostic methods, such as dipstick tests and microscopy, are often inaccurate and require trained personnel and specialised equipment, making them difficult to implement in low-resource settings. Additionally, these methods are often time-consuming and may not provide results in real-time, which can delay diagnosis and treatment.

The development of more affordable and easy-to-use diagnostic solutions could significantly improve access to urinalysis and enable the earlier detection and diagnosis of diseases. An example is using electrochemical biosensors to detect changes in the electrical current that occurs when a specific biomolecule binds to the sensor. Combining biosensors with microfluidics can be beneficial, since this technology can use small droplets or volumes of liquid samples, allowing for the highly sensitive and specific detection of biomarkers in urine.

Combining an increasing knowledge of the pathophysiology of a number of diseases with the development of portable, non-invasive, and highly sensitive devices for point-of-care diagnostics, one of the most active research areas in microfluidic biosensors for urinalysis could allow for the rapid detection of even very low levels of biomarkers in urine. This can enable, for instance, an ever earlier diagnosis of various types of cancer, and more precise monitoring of health status and chronic diseases. However, future validation of reference concentrations of certain biomarkers in urine is still required.

Implementing a biosensor combined with microfluidics in a smart toilet could revolutionise point-of-care urinalysis. The smart toilet would be equipped with a biosensor that can detect and analyse a wide range of biomarkers in urine, providing diagnostic information about the user’s health. Furthermore, the smart toilet could continuously monitor the user’s health and provide diagnostic information on an ongoing basis, with the data collected by the toilet transmitted wirelessly to a remote monitoring system, allowing healthcare professionals to access the data in real-time.

## Figures and Tables

**Figure 1 biomedicines-11-01051-f001:**
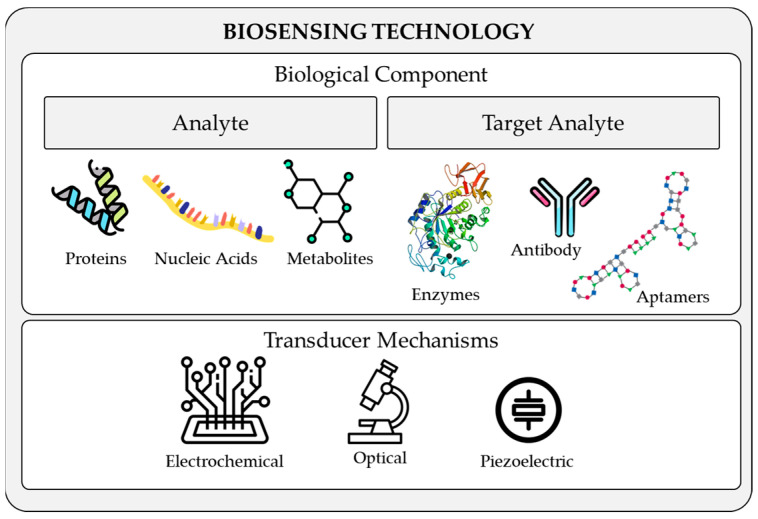
Schematic of different parts of a biosensor.

**Figure 2 biomedicines-11-01051-f002:**
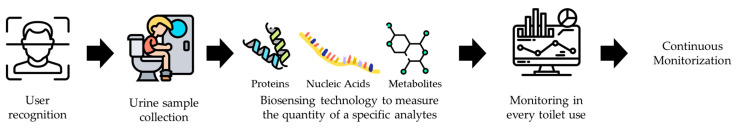
Schematic to demonstrate the process of continuous monitoring using a toilet to collect the sample.

**Table 2 biomedicines-11-01051-t002:** Summary of the transducer mechanisms used in biosensing and their advantages and disadvantages.

Transducer	Sensing Mechanism	Advantages	Disadvantages
Electrochemical	Measures changes in electrical properties, such as voltage, current, or impedance, resulting from the interaction between the target analyte and the sensing electrode.	Ease of use Low cost High sensitivity Low power requirements Low sample volume	Sensitive to the surrounding environment Sensitive to pH, temperature, and storage conditions.
Optical	Utilizes light as the sensing mechanism, either by measuring the absorption, fluorescence, or scattering of light by the target analyte.	Low detection limit Versatility Real-time detection Low sample volume	Sensitive to the surrounding environment Sensitive to pH, temperature, and light.
Piezoelectric	Measures the changes in mass or viscosity of the target analyte by detecting the mechanical vibrations generated by the interaction between the analyte and the sensing crystal.	High sensitivity Versatility Low detection limit Real-time detection Low sample volume	High cost Fragility Temperature-dependent sensitivity

**Table 3 biomedicines-11-01051-t003:** Summary of the advantages and disadvantages of the different materials used in the fabrication of microfluidic structures and systems.

Material		Advantages	Disadvantages
Silicon		Thermostability Design flexibility Chemical compatibility Semiconducting properties	Opacity Expensive High elastic modulus
Glass		Thermostability Optical transparency Biologically compatible High resolution at the μm scale	Microfabrication difficulties Time-consuming labor Preparation in cleanrooms
Polymers	PMMA	Inexpensive Optical transparency Good mechanical properties Allows surface modification	Sensitive to scratches Poor resistance to many chemicals Dissolves in many solvents
	PDMS	Inexpensive Gas permeability Rapid prototyping Optical transparency	Incompatible with organic solvents Low mechanical strength Unstable surface treatments
Paper		Low cost Accessibility Biocompatibility High physical absorption	Thickness requirements for achieving transparency Poor mechanical strength in a wet state

**Table 4 biomedicines-11-01051-t004:** Summary of the advantages and disadvantages of the different microfluidics fabrication techniques, adapted from the literature [153,154].

Fabrication Technique	Advantages	Disadvantages
Injection Moulding	Easy to fabricate complex geometries 3D geometries Low cycle time Mass production Highly automated	Material restriction Mould features must not have undercuts Expensive fabrication Limited resolution
Hot Embossing	Cost-effective Precise Rapid replication of microstructures Mass production	Material restriction Difficult fabrication of complex 3D geometries
Photolithography	High wafer throughputs Ideal for microscale features High resolution (down to a few nm)	Requires a flat surface to start Chemical post-treatment needed Need of cleanroom facilities
Soft Lithography	Cost-effective 3D geometries High resolution (down to a few nm)	Pattern deformation Vulnerable to defect Need of cleanroom facilities

## Data Availability

Not applicable.

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
