# Peer review of "Urinary Biomarkers and Point-of-Care Urinalysis Devices for Early Diagnosis and Management of Disease: A Review"

_biomedicines, 2023, doi:10.3390/biomedicines11041051_

Round 1

Reviewer 1 Report

In the forts part of the paper some examples of biomarkers for different disease conditions are reported and associated with so-called metabolomic approach.

Biosensors and microfluidic technologies are well described and summarized in the tables. 

However, it is not readily evident which are possible applications in the routine analysis, even as POCT instruments. Moreover, the use of these devices in the continuous monitoring is somewhat confusing and it should be better defined the difference between static and dynamic measurements. 

Reviewer 2 Report

This manuscript presents a comprehensive review of the current state of biosensing and microfluidics technologies for point-of-care urinalysis. The review first discusses biomarkers that can be used for the diagnosis and monitoring of a variety of diseases, and then along with different techniques and materials for the fabrication of biosensing technologies and microfluidic structures in point-of-care urinalysis. And finally, concludes and emphasizes the need for more affordable and easy-to-use diagnostic solutions for urinalysis and highlights the potential of combining biosensors with microfluidics in a smart toilet for continuous monitoring of biomarkers.

Overall, the manuscript is well-written and presents good logic. However, as a review paper focusing on urine biomarkers, biosensing technologies, and microfluidic devices, the authors still need some additional work to cover a broader range of urine biomarkers other than metabolites, adding more information about different biosensing techniques in urinalysis, and elaborate more about how microfluidics are uniquely applied in urinalysis compared to other sample types and its potential to be used at point-of-care or continuous monitoring. The advantage of microfluidic biosensing technologies over the current gold standard urine test (e.g. dipstick test or lateral flow assay) needs to be compared and discussed. Some figures and schematics are suggested to be added to this paper. I would recommend that the authors revise the manuscript to address the following comments and that the revised manuscript undergo another round of peer review.

Major Comments:

1.      Despite urine being a rich source of cellular metabolites, other larger molecules like proteins, DNA/RNAs should not be neglected by this review. For example, urine albumin is one of the key proteins that is often used as a biomarker for kidney function and damage and should not be characterized under the umbrella term of metabolite. Other biomarkers mentioned in the review like urinary bladder cancer antigen, PSA, NGAL, amyloid ?, are all should be considered protein biomarkers. Other proteins that are closely related to inflammations are cytokines, chemokines, and growth factors.

For example

a.       Prasad, S., Tyagi, A. K., & Aggarwal, B. B. (2016). Detection of inflammatory biomarkers in saliva and urine: potential in diagnosis, prevention, and treatment for chronic diseases. Experimental Biology and Medicine, 241(8), 783-799.

b.      Liu, B. C., Zhang, L., Lv, L. L., Wang, Y. L., Liu, D. G., & Zhang, X. L. (2006). Application of antibody array technology in the analysis of urinary cytokine profiles in patients with chronic kidney disease. American journal of nephrology, 26(5), 483-490.

Therefore, the authors should consider organizing a separate discussion section for protein analysis or proteomics. On the other hand, while urine may not be the most ideal sample type for nucleic acid detection, there are still quite a few studies that have investigated the use of urine mRNA and DNA as biomarkers for various diseases and conditions, including cancer, infectious diseases, and kidney disease.

2.      It is great that the authors organized urine biomarkers based on the disease type. It will be even better if the authors could also provide the typical concentrations of those biomarkers and whether it is challenging to use urine as the sample source to detect them and whether some ultrasensitive biosensing or microfluidic technologies (e.g. single molecule counting assays) could address those issues.

3.      As mentioned in comment 2, in the biosensing and microfluidics technologies sections, their uniqueness or challenge in urine biomarker analysis should be further evaluated and elaborated. These two sections may also need some figures and schematics to be more informative. Right now, these two sections are a simple overview of general biosensors and microfluidics. And the advantages of those technologies over the current gold standard in urinalysis could be also discussed.

4.      It will be great to add a figure in the Point-of-Care Diagnostics: Urinalysis section as well, for example showing some schematic concepts of those smart toilets and cartridges for readers to visualize some continuous monitoring system.

Minor Comments:

1.      There are some typographical errors throughout the manuscript that should be corrected before publication. For example, the “PDMS” in table 2 is spelled as “PMDS”. 

Reviewer 3 Report

“Current diagnosis of kidney cancer occurs when symptoms appear, and when the prog-     271
nosis is very poor, so alternative diagnosis methods that allow for its detection at early         272
stages are needed

I would recommend changing this sentence. Today, most kidney cancers are detected before symptoms appear due to the widespread use of imaging modalities mainly US for a number of different conditions such as routine/systemic examinations or back pain (caused by the spine) or abdominal pain due to gastrointestinal diseases, these are called incidental kidney cancer, this led to a significant reduction in the stage of the disease. Of course, this does not mean that new markers, especially those from urine, are not needed.

Round 2

Reviewer 2 Report

The reviewer's comments have been addressed by the authors and they have made significant expansions to the types of the urine biomarkers. No further comments are added.